# Innovation-Based Fault Detection and Exclusion Applied to Ultra-WideBand Augmented Urban GNSS Navigation

**Paul Zabalegui** [1,2,*] , **Gorka De Miguel** [1,2] , **Jaizki Mendizabal** [1,2] and **Iñigo Adin** [1,2]

1 CEIT-Basque Research and Technology Alliance (BRTA), 20018 Donostia-San Sebastián, Spain
2 Electronic Engineering Department, Universidad de Navarra, Tecnun, 20018 Donostia-San Sebastián, Spain
* Correspondence: pzabalegui@ceit.es

**Abstract:** Due to their ability to provide a worldwide absolute outdoor positioning, Global Navigation Satellite Systems (GNSS) have become a reference technology in terms of navigation technologies. Transportation-related sectors make use of this technology in order to obtain a position, velocity, and time solution for different outdoor tasks and applications. However, the performance of GNSS-based navigation is degraded when employed in urban areas in which satellite visibility is not good enough or nonexistent, as the ranging signals become obstructed or reflected by any of the numerous surrounding objects. For these situations, Ultra-Wideband (UWB) technology is a perfect candidate to complement GNSS as a navigation solution, as its anchor trilateration-based radiofrequency positioning resembles GNSS's principle. Nevertheless, this fusion is vulnerable to interferences affecting both systems, since multiple signal-degrading error sources can be found in urban environments. Moreover, an inadequate location of the augmenting UWB transmitters can introduce additional errors to the system due to its vulnerability to the multipath effect. Therefore, the misbehavior of an augmentation system could lead to unexpected and critical faults instead of improving the performance of the standalone GNSS. Accordingly, this research work presents the performance improvement caused by the application of Fault Detection and Exclusion methods when applied to a UWB-augmented low-cost GNSS system in urban environments.

**Keywords:** Ultra-Wideband; GNSS augmentation; Kalman-Filtering; urban navigation; Intelligent Transport Systems; Fault Detection and Exclusion (FDE)



## 1. Introduction

The use of positioning methods has become a common practice among the different means of transport to monitor and enhance different aspects of vehicles, such as the efficiency of the driving, the route, safety, or location of freight [1,2]. Of all the available options, GNSS-based positioning has become the most employed technology as it is based on an already deployed infrastructure that allows the user to locate any object with just a receiver, this being an easy and scalable low-cost option. This technology offers global positioning to every receiver that has a minimum of four satellites in its range of visibility. When navigating through urban environments, the surrounding objects such as buildings may block the Line-of-Sight (LOS) signal coming from a low-elevation satellite, leading to a reception based on just Non-Line-of-Sight signal components, which can result in a poor position estimation [3–6]. Consequently, and due to the fact that the signals received from low-elevation satellites have lower Signal-to-Noise-Ratio (SNR) and are more vulnerable to atmospheric effects, the elevation masks are usually increased to filter all satellites with an a priori poor-quality signal. This fact leads to a higher chance of encountering the unavailability of GNSS navigation due to low accuracies or the lack of continuity caused by the absence of the required minimum number of satellites in indoor environments.

The multisensor navigation approach is a common option to face this problem. In the last decades, sensors such as inertial measurement units (IMU), barometers, odometers,

magnetometers and digital compasses have been used. However, according to the author in [7], recent trends have focused on the use of image-based, terrain-based collaborative navigation. They also stated that the choice of sensors should be aligned with the characteristics of the use case and the target application. Similarly, the author in [8] affirmed that this choice should take into account the environment, dynamics, budget, accuracy requirements, and the degree of robustness or integrity required. As an example, it discusses the use of odometers, magnetic compasses, barometers, and map-matching algorithms as a typical augmentation in road vehicles, while trains may also use Doppler radar. Furthermore, it mentions that cell phone, UWB, and WLAN positioning may supplement GNSS indoors and in urban areas.

In fact, the use of Ultra-Wideband (UWB) radio technology in urban environments is a reliable option when trying to solve the main drawback of GNSS in low-visibility scenarios, indoor environments and even outdoor-indoor transitions, as the sub-meter accuracy and the tailored transmitting anchor geometry provide reliable information source at the time of positioning vehicles, tracking assets in warehouses, improving productivity in assembly lines, etc. [9–11]. Previous research shows that, as the UWB technology is analogous to GNSS as the positioning is based on the trilateration of the signals received from transmitting anchors, the two technologies can be fused using different algorithms and strategies [12–15]. This fact could allow deploying UWB transmitting anchors in low-visibility areas, such as urban canyons, underground garages, or tunnels (see Figure 1), in order to not lose the continuity of the positioning and even improve GNSS' accuracy in required areas.

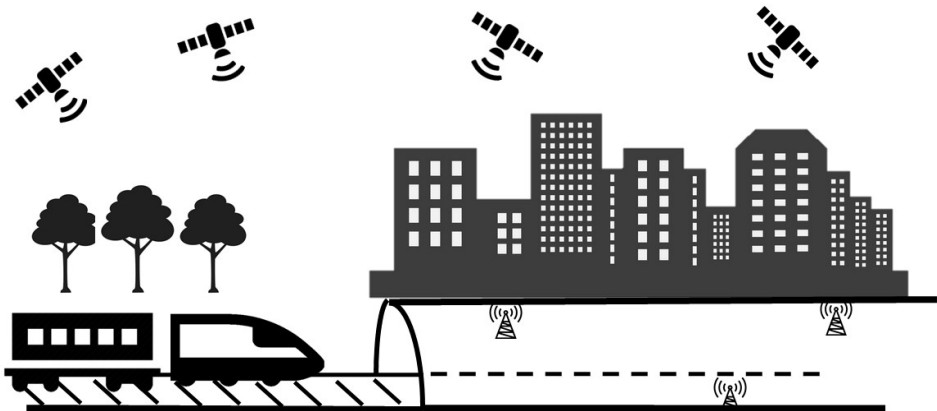

**Figure 1.** Use case of the UWB-aided GNSS in a tunnel [16].

In spite of the improvement in performance that UWB can imply to GNSS, it is still a radio frequency-based technology and, consequently, it is also vulnerable to interferences or error sources such as multipath, electromagnetic noise, and jamming [17–19]. These error sources are the same ones that degrade the signal reception in the GNSS receivers and cause a bad estimation of the ranges between receivers and transmitting nodes such as anchors or satellites, which leads to a bad position solution. Moreover, due to the vulnerability of UWB signals to multipath, inappropriate UWB anchor geometries or anchor locations may induce the degradation of certain signals as a result of the multipath effect.

Fault Detection and Exclusion methods were created as an extension to the Receiver Autonomous Integrity Monitoring (RAIM) methods that were developed for navigating in the aviation domain. These methods were developed to perform an autonomous self-check of the integrity of each and every GNSS observable and exclude the ones that are assumed to be faulty. In this document, the continuation of the research work discussed in [16] is presented in Section 2. As shown in [16], the Fault Detection and Exclusion (FDE) methods originally developed for GNSS-based navigation can be utilized for the improvement of UWB-based navigation. In the current piece of work, the performance-improving effect of

said methods is shown when applied to the innovation vector of a Kalman Filter when this is used to employ UWB signals as an augmentation for GNSS navigation. The main purpose, accordingly, is to prove that FDE methods can be used to exclude faulty augmentation observations since the misbehavior of said augmentation systems could lead to unexpected and critical faults instead of improving the performance of the standalone GNSS.

For said purpose, a theoretical approach to the performed research is presented in Section 2. To do so, the Kalman Filter-based fusion is first presented, in order to, then, discuss the employed FDE method. Afterwards, the description of the measurement site is introduced in Section 3, only to then show and analyze the results of applying this FDE method to the UWB-augmented fusion algorithm in Section 4. Finally, the obtained conclusions are presented in Section 5.

## 2. Theoretical Approach

The following section presents a theoretical approach to the employed navigation algorithm, known as the Kalman Filter. Moreover, the fusion between GNSS and UWB will be discussed. Afterwards, the Kalman Filter-based reliability testing is explained, in order to then explain the employed FDE scheme.

### 2.1. Kalman Filtering-Based GNSS and UWB Fusion

The Kalman Filter is the basis of most of the estimation algorithms used in navigation, as it is employed for different purposes such as smoothing navigation solutions and even fusing GNSS data with other navigation sensors. Its main characteristic is that it is able to maintain real-time estimates of continuously changing system parameters like position and velocity. This is carried out using both deterministic and statistical properties of the mentioned parameters and making assumptions about the input measurements and their characteristics and uncertainties. Moreover, due to the recursive nature of this algorithm, it employs past measurements together with a dynamic model of the navigation system for the aim of achieving a more accurate PVT estimation.

In this research work, the nonlinear version of the Kalman Filter was employed, also known as an Extended Kalman Filter (EKF), which linearizes an estimate of the current mean and covariance [20]. This algorithm is usually initialized with a time-invariant parameter x that has been computed employing a snapshot algorithm such as the LSE method, for example. It is then assumed that this parameter is not only time-variant but that it can be described by a dynamic model that relates two adjacent epochs as

$$\hat{x}_k = \Phi_{k-1}\, x_{k-1} + w_k,\ w_k \sim N\big(0,\ Q_{w_k}\big), \tag{1}$$

where $\hat{x}_k$ is the estimated parameter; $\Phi_{k-1}$ describes the dynamic model-dependent transition matrix, which defines the change of the state vector with time as a function of the system's dynamics; $x_{k-1}$ describes the last known value of the parameter to estimate; $w_k$ models the system process noise, and $Q_{w_k}$ is the covariance matrix of the process noise.

For this research work, the transition matrix, $\Phi_{k-1}$, has been modelled following a constant velocity dynamical assumption (see [20] for more information about the dynamical modeling of a system). Consequently, the employed state vector if formed as

$$\hat{x}_k = \begin{bmatrix} pos_{\text{ECEF}_x} \\ pos_{\text{ECEF}_y} \\ pos_{\text{ECEF}_z} \\ vel_{\text{ECEF}_x} \\ vel_{\text{ECEF}_y} \\ vel_{\text{ECEF}_z} \\ \delta t \\ \dot{\delta t} \end{bmatrix} \tag{2}$$

where $\left(pos_{ECEF_x}, pos_{ECEF_y}, pos_{ECEF_z}\right)$ describe the computed position, $\left(vel_{ECEF_x}, vel_{ECEF_y}, vel_{ECEF_z}\right)$ describe the computed velocity of the rover and $\delta t$, $\dot{\delta t}$ describe the receiver's clock offset and clock drift, accordingly.

This estimation of parameter $\hat{x}_k$, nevertheless, comes from an uncertain method as the LSE is and, consequently, the uncertainty about this estimation is inherited by the future values. It is, thus, necessary to propagate said uncertainty, so that its unbounded propagation degrades the results. Of the numerous forms of covariance propagation in use, one of the most common is the following one:

$$\hat{P}_k = \Phi_{k-1} P_{k-1} \Phi_{k-1}^{\mathsf{T}} + Q_{k-1}, \tag{3}$$

where $\hat{P}_k$ is the propagated covariance matrix; $\Phi_{k-1}$ describes the dynamic model-dependent transition matrix; $P_{k-1}$ describes the a priori covariance matrix, and $Q_{k-1}$ represents the covariance matrix of the system noise, which defines the increase with time of the uncertainties of the state estimates (i.e., unmeasured dynamics and instrument noise).

The KF does not only trust the quality of the input state or the propagated estate. Instead of this, it determines a weight of measure of trust for the measured information and for the estimated state in order to combine both of them, avoiding poor estimations and observation outliers. This is carried out by means of the Kalman Gain matrix, which is computed as

$$\hat{K}_k = \hat{P}_k H_K^T \left( H_k \hat{P}_k H_K^T + R_k \right)^{-1} = \hat{P}_k H_K^T S_k^{-1}, \tag{4}$$

where $\hat{K}_k$ is the computed Kalman Gain matrix; $H_k$ represents the measurement matrix, which defines how the measurement vector varies with the state vector; $R_k$ describes the measurement noise covariance matrix and $S_k$ models the de innovation vector's covariance matrix. Note that the employed measurement matrix was formed as shown in [16], but it was expanded in order to fit UWB observables. This matrix is formed as follows:

$$H = \begin{bmatrix} a_{x_{GNSS}1} & a_{y_{GNSS}1} & a_{z_{GNSS}1} & 0 & 0 & 0 & 1 & 0 \\ a_{x_{GNSS}2} & a_{y_{GNSS}2} & a_{z_{GNSS}2} & 0 & 0 & 0 & 1 & 0 \\ \vdots & \vdots & \vdots & \vdots & \vdots & \vdots & \vdots & \vdots \\ a_{x_{GNSS}n} & a_{y_{GNSS}n} & a_{z_{GNSS}n} & 0 & 0 & 0 & 1 & 0 \\ a_{x_{UWB}1} & a_{y_{UWB}} & a_{z_{UWB}1} & 0 & 0 & 0 & 0 & 0 \\ a_{x_{UWB}2} & a_{y_{UWB}2} & a_{z_{UWB}2} & 0 & 0 & 0 & 0 & 0 \\ \vdots & \vdots & \vdots & \vdots & \vdots & \vdots & \vdots & \vdots \\ a_{x_{UWB}m} & a_{y_{UWB}m} & a_{z_{UWB}m} & 0 & 0 & 0 & 0 & 0 \\ 0 & 0 & 0 & a_{x_{GNSS}1} & a_{y_{GNSS}1} & a_{z_{GNSS}1} & 0 & 1 \\ 0 & 0 & 0 & a_{x_{GNSS}2} & a_{y_{GNSS}2} & a_{z_{GNSS}2} & 0 & 1 \\ \vdots & \vdots & \vdots & \vdots & \vdots & \vdots & \vdots & \vdots \\ 0 & 0 & 0 & a_{x_{GNSS}n} & a_{y_{GNSS}n} & a_{z_{GNSS}n} & 0 & 1 \end{bmatrix} \tag{5}$$

where $a_{xi}$, $a_{yi}$ and $a_{zi}$ (both for GNSS and UWB) represent the unitary vectors (in ECEF x-y-z coordinates) that link the receiver's position and each of the transmitting satellite/anchor's position. This weighting matrix is applied, as mentioned, as a measure of trust that is given to the measurements and the estimations. Having computed the mentioned parameters, this yields

$$x_k = \hat{x}_k + \hat{K}_k(z_k - H_k \hat{x}_k) = \hat{x}_k + \hat{K}_k \delta z_k, \tag{6}$$

where $x_k$ represents the computed state; $z_k$ corresponds to the read measurement vector; and $\delta z_k$ represents the innovation vector, which describes the difference between the read measurement and the expected measurement.

Due to the similarity of the measurements between GNSS and UWB, the fusion between these two can be intuitively performed by adding the UWB ranges to the observable vector, $z_k$, this being

$$
z_k = \begin{bmatrix} \Delta\rho_1 \\ \vdots \\ \Delta\rho_n \\ \Delta r_1 \\ \vdots \\ \Delta r_m \\ \Delta\dot{\rho}_1 \\ \vdots \\ \Delta\dot{\rho}_n \end{bmatrix} = \begin{bmatrix} \rho_1 - \hat{\rho}_1 \\ \vdots \\ \rho_n - \hat{\rho}_n \\ r_1 - \hat{r}_1 \\ \vdots \\ r_m - \hat{r}_m \\ \dot{\rho}_1 - \hat{\dot{\rho}}_1 \\ \vdots \\ \dot{\rho}_n - \hat{\dot{\rho}}_n \end{bmatrix},
\tag{7}
$$

where $\rho_i$ is the pseudorange measurement for the $i_{th}$ satellite; $r_j$ represents the range measurement for the $j_{th}$ anchor; $\dot{\rho}_i$ contains the Doppler measurement for the $i_{th}$ satellite; $n$ denotes the number of visible GNSS satellites and the number of observable pairs; and $m$ is the number of visible UWB anchor measurements. Another design parameter of the Kalman Filter that depends on the number of measurements and their characteristics is the measurement covariance matrix described by $R_k$. In this research, the elements of this diagonal matrix were weighted according to the power spectral density (PSD) and reweighted, then, according to the signal SNR and satellite elevation in the case of the satellite signals and according to the elevation in the case of the UWB signals. The diagonal elements of the matrix that correspond to satellite signals are characterized as

$$
\sigma_{sat_{ii}} = f(PSD_{GNSS})\, f(elevation_{sat_{ii}},\, SNR_{sat_{ii}}),
\tag{8}
$$

while the diagonal elements that correspond to UWB signals are characterized as

$$
\sigma_{Anchor_{jj}} = f(PSD_{UWB})\, f\left(elevation_{anchor_{jj}}\right),
\tag{9}
$$

The fusion of these two types of measurements is based on the assumption that UWB range estimations have significantly higher accuracy than the ones provided by a low-cost GNSS receiver. Accordingly, even if the estimation of the range of UWB observables would get slightly degraded, these should still be valid as an augmentation source, since the range estimation error would still be better than the one corresponding to GNSS. Consequently, slight errors in the location of the anchors or slight errors in the synchronization of said ranges with the ones incoming from the GNSS receiver should still improve the performance of the standalone GNSS.

*2.2. Reliability Testing*

As can be deduced from (6), the existence of an outlier in the measurement vector, $z_k$, may induce an error in the computed result, $x_k$. Thus, testing the internal consistency of the observations, which is to say, detecting the presence of any bias or outlier, turns out to be necessary if ensuring the reliability of the solution is pursued. Accordingly, a wide variety of research lines have studied fault diagnosis and prognosis functions using system theory and statistical decision theory [21–23]. In this research work, similarly to what was presented in [16,23], a test statistic was introduced in order to estimate or quantify the consistency of the measurements. This parameter was obtained from the normalized square sum of the innovation vector, $\delta z_k$, of the Kalman Filter, which can be computed as

$$
w_k = \delta z_k^T\, S_K^{-1}\, \delta z_k,\ w_k \sim X^2(p,\lambda),\ p \in N,\ \lambda \in \mathbb{R}_{>0},
\tag{10}
$$

This test statistic, $w_k$, follows a central chi-square distribution (i.e., $w_k \sim X^2(p,0)$) with $p$ degrees of freedom when the observation errors are distributed according to a zero-mean

normal distribution (i.e., $e \sim N(0, \Sigma)$). On the other hand, when there are bias errors along with the measurements (i.e., $e \sim N(\mu, \Sigma)$), the test statistic follows a non-central chi-square distribution with a centrality parameter $\lambda_0$ (i.e., $w_k \sim X^2(p, \lambda_0)$). This parameter is used to compare its value against a certain threshold so that the following hypotheses can be verified

**Hypothesis H$_0$.** *No fault is present in the observations (Nominal condition).*

**Hypothesis H$_a$.** *A fault is present in the observations (Fault condition).*

According to these hypotheses, two decisions can be taken: rejecting the measurement or accepting it (see Table 1). Therefore, four different scenarios can take place

**Table 1.** Fault hypotheses and actions [14].

|  | **H$_0$ Accepted** | **H$_0$ Rejected** |
|---|---|---|
| H$_a$ false | Correct decision | False alarm |
| H$_a$ true | Missed detection | Correct rejection |

Therefore, the false alarm concept is defined as an indication of positioning failure when this has not occurred. In a similar way, a missed detection event is defined as an indication of positioning failure that has not been detected, which must be minimized as it can lead to fatal events. These hypotheses are tested by comparing the test statistic computed as (10) and a threshold that depends on the probability of false alarm and the degrees of freedom of the distribution, which can be denoted as $X^2_{1-P_{fa}, p}$, being $P_{fa}$ the false alarm rate and $p$ the degrees of freedom. This comparison is known as Global Test (GT) and is used to detect anomalies in the measurements. According to this, as seen in Figure 2, there is a user-defined frontier value of $T_D$ that intersects both curves, which acts as a delimiter to choose between the curves. This threshold value, then, delimits the faulty and non-faulty measurements. In other words, if the test statistic falls below this threshold value, it will be considered a non-faulty observation, while if it falls above the threshold, it will be considered a faulty observation. Note that as the curves in Figure 2 intersect, there are values of the non-faulty distribution that will be considered as faulty and values of the faulty distribution that will be thought of as non-faulty ones. The former case's probability will be denoted by the probability of a false alarm ($P_{fa}$), while the latter one will be called the probability of missed detection ($P_{md}$). These values can be computed from the graphs integrating the corresponding probability density function (pdf) from zero to the threshold in the case of $P_{md}$ and from the threshold to infinity in the case of $P_{fa}$. The threshold, $T_D$, can be computed as the inverse of a chi-square distribution with $n - p$ degrees of freedom and probability of false alarm $P_{fa}$:

$$X^2_{1-P_{fa}, p} = T_D \tag{11}$$

This statistical behavior is the base of the Global Test that measures the reliability of the observations. This test is used to estimate the scenario hypothesis and, if necessary, to proceed to a Local Test with more specific alternative hypotheses for failure isolation.

The outlier detection and isolation are based on assumptions according to estimated observational residuals. These residuals, nevertheless, are just indicatives of both the behavior of a mathematical model and the observations. As a consequence, the differentiation of both is not easy, as a bad geometrical model and model assumptions or bad observations affect the residuals in the same way [24]. However, the author in [24] states that the most likely reason for the rejection of the null hypothesis is the existence of outliers and their corresponding detection during the global test. In these processes, an assumption of Gaussian zero-mean noise is made for the unbiased error-free case of the linearized model.

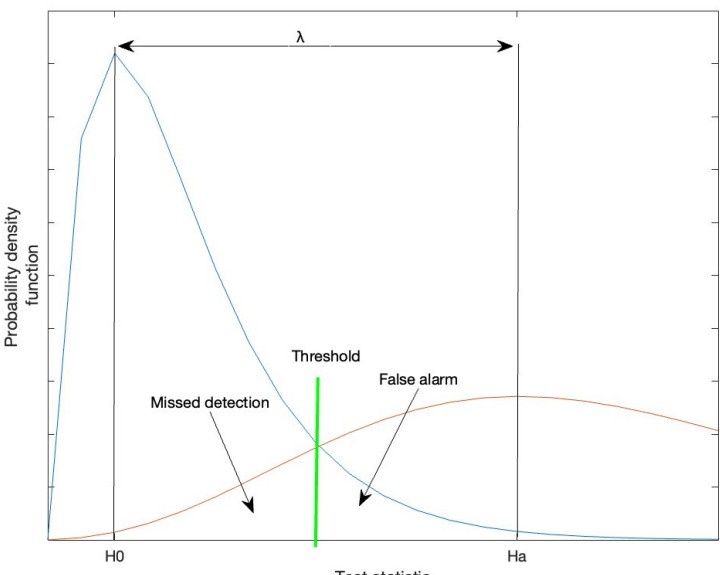

**Figure 2.** Central and non-central chi-square distributions used in the global test.

Therefore, failure isolation may not be as straightforward as expected, since matrix $S_k$ involves the cross-correlations of the state components. Consequently, the effect of the faulty measurement could leak into the rest of the observables, leading to mistaken error exclusions. In order to solve such an issue, the author in [25] proposed normalizing the innovation vector by the Cholesky decomposition of matrix $S_k$, which results in a positive semi-definite matrix with no more correlations between its elements, as these are removed. This can be denoted as

$$S_k^{-1} = M_k^T M_k,$$
$$\hat{\delta z}_k = M_k \delta z_k \tag{12}$$

where $\hat{\delta z}_k$ represents the uncorrelated normalized innovation vector and $M_k$ denotes the positive semidefinite uncorrelated covariance matrix, which is the outcome of the Cholesky decomposition.

Having normalized the innovation vector, it is assumed that the element with the highest value is the one to be isolated. Note that this approach is compatible with an iterative scheme, which will be introduced in the following Section 2.3.

### 2.3. Fault Detection and Exclusion Scheme

As discussed in previous research [16], the classical FDE method was originally developed as an extension of the classical Receiver Autonomous Integrity Monitoring (RAIM) for GNSS and was designed under a single failure assumption, due to the friendly radiofrequency-related conditions available in the sky. This single failure assumption could have been enough for the aviation domain, where the probability of observable failure was much lower than the systemic failure of GNSS, which implied a single execution of the Global test—Local test pair. Nevertheless, this may not be enough for an urban scenario, where the probability of finding a faulty observable is significantly higher than in open scenarios (i.e, the sky). Moreover, the likelihood of finding faulty observables increases when $2n + m$ observables are used due to the combination of GNSS and UWB technologies.

Consequently, the classical FDE scheme can be conveniently adapted to an iterative scheme (see Figure 3) in which the global and local test combination is executed until no more observable failures are detected or there is no longer enough observable redundancy. Intuitively, and due to the high confidence that is given to the UWB observables as they act as an augmentation system, UWB range measurements turn out to be the most probable values to be excluded first under the existence of a faulty UWB measurement.

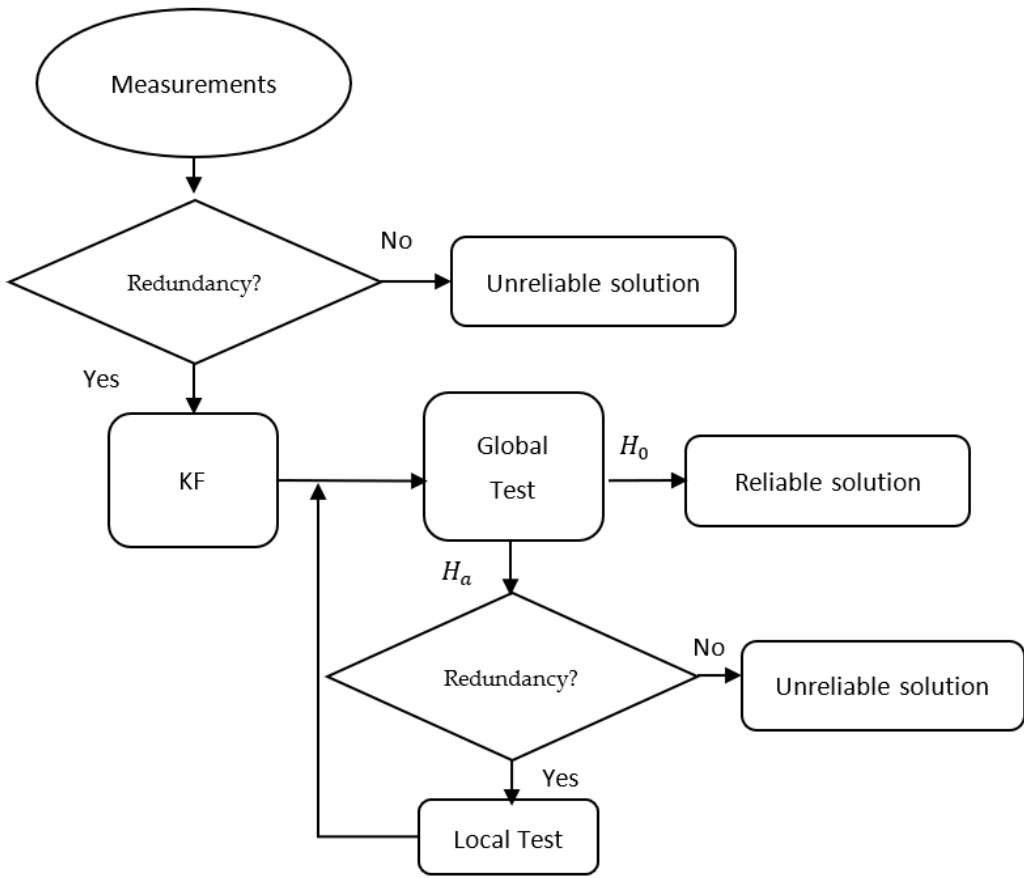

**Figure 3.** Block diagram of the employed iterative FDE method.

Note that other FDE schemes such as the ones discussed in [24,26–29] could also be used, just by adapting the local test to the condition discussed in the previous subsection. Depending on the number of augmentation UWB anchors, nevertheless, the computation time could drastically increase, as commented in [16].

## 3. Description of the Measurement Campaign

The data used in the validation of this paper were obtained in the measurement campaign explained in this section.

### 3.1. Test Scenario

This measurement scenario is located at the technological park of Miramon, San Sebastian, Spain, where a closed-shaped track with indoor and outdoor sections was selected for the campaign (Figure 4). This suburban site is a suitable environment to check the performance of the navigation system, as it contains four different environments with different characteristics within the same measurement: (1) a good-visibility open-sky part, (2) a low-building-density part, (3) a complex urban canyon at the entrance of the garage (Figure 5), and (4) an indoor part inside the garage (Figure 5). The combination of the mentioned areas in the same measurement track allows the analysis of the employed FDE method when applied to sites with different characteristics. Having said this, this study focused on the behavior of the FDE method when applied to UWB-augmented GNSS navigation in urban and indoor environments. Accordingly, the UWB anchors have only been located at the entrance of the garage and in the indoor environment inside the garage.

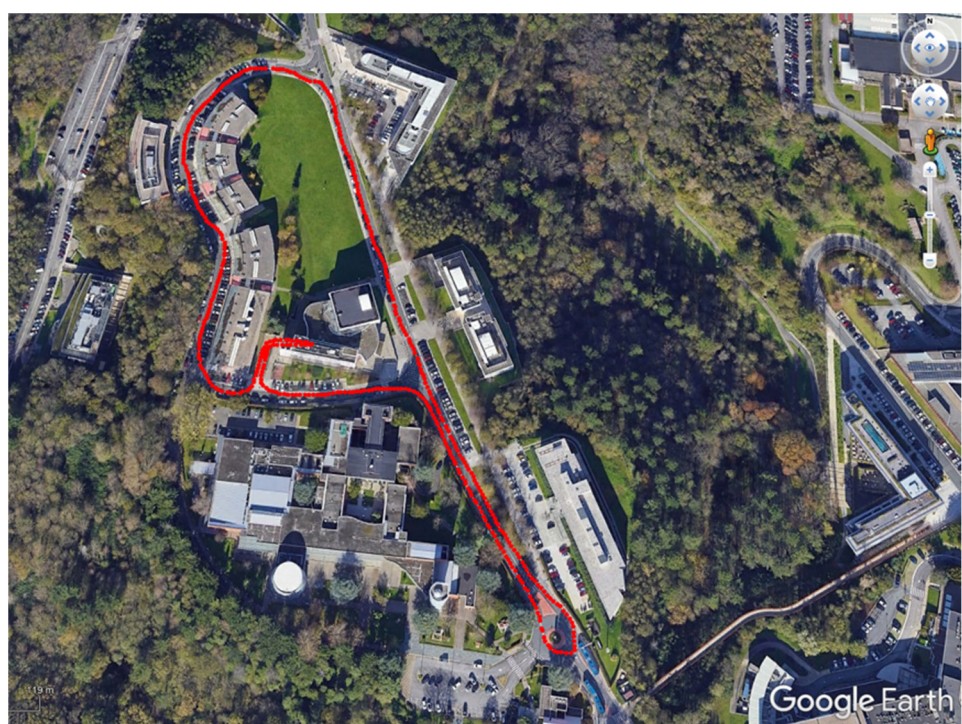

**Figure 4.** Complete view of the measurement site and performed trajectory.

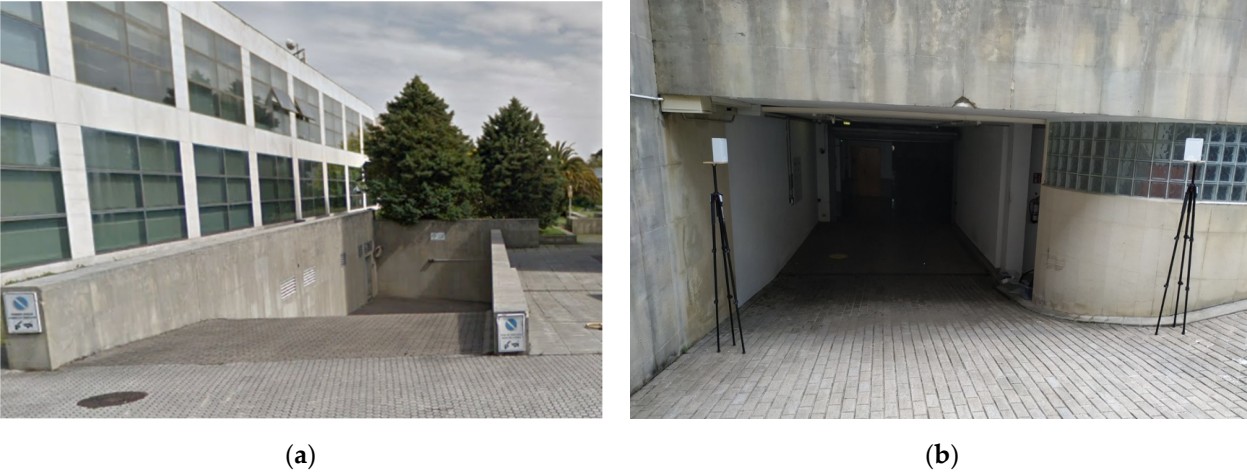

    (**a**)                                                 (**b**)

**Figure 5.** Example pictures of the harsh environment: (**a**) beginning of the urban canyon and (**b**) beginning of the indoor environment.

### 3.2. Test Setup

The employed navigation equipment was composed of low-cost devices, as the objective of this research work was to improve the performance of UWB-augmented low-cost GNSS receivers by employing fault-excluding methods based on the statistical behavior of these. This way, the input measurements were provided by an Ublox M8T GNSS receiver [30] and proprietary UWB tag and anchors that contained the DW1000 UWB transceiver [31]. The choice of these devices was mainly based on their low cost and their packages, together with the following specifications (Table 2).

**Table 2.** Characteristics of the employed hardware.

| Devices | Ublox M8T | Decawave DW1000 |
|---|---|---|
| Power supply | 2.7 V to 3.6 V <br> 28 mA @ 3.0 V | 2.8 to 3.6 V <br> 64 mA |
| Dimensions | 17.0 × 22.4 × 2.4 mm | 6 × 6 × 2.4 mm |
| Range accuracy [m] | 2.0 | <0.1 |
| Others | Multiconstellation | Supports 6 RF channels from 3.5 to 6.5 GHz |

This set-up of the equipment was carried out in the following way: the corresponding receivers, the reference high-performance GNSS Septentrio system, and the recording computer were located inside in the rack shown in Figure 6, while the GNSS antenna and the UWB tag were installed on the roof of a car (see Figure 7).

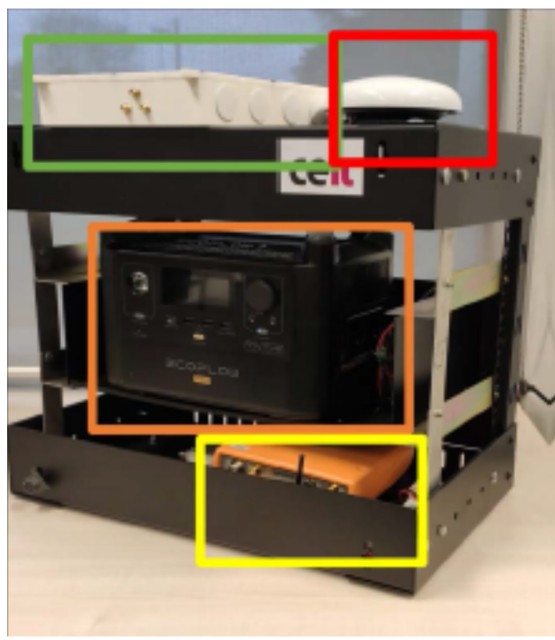

**Figure 6.** Employed measurement system (red: GNSS antenna; green: navigation system; orange: power supply; yellow: ground truth generating system).

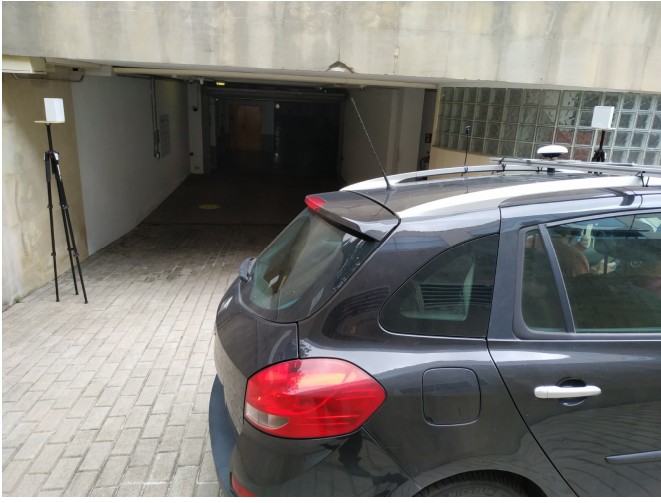

**Figure 7.** The starting point of the measurement lap at the entrance of the garage and the location of the employed GNSS antenna and UWB tag on the roof of the car.

Moreover, when configuring the UWB devices, an asynchronous ranging was used due to the absence of clock synchronization. Because a higher rate (10 Hz) was configured in UWB devices than in the GNSS receiver (1 Hz), the age of UWB signals cannot exceed the GNSS epoch in more than 100 ms. Since the velocity of the rover did not surpass 15 km/h during the manoeuvres inside the UWB coverage area, this synchronization error would not exceed 0.4 m. According to the assumption mentioned at the end of Section 2.1, since the range estimation error of GNSS receivers is higher than this maximum synchronization error, this should not degrade the performance of the fusion, allowing the use of UWB signals as an augmentation method.

Moreover, eight UWB anchors were placed within the limits of the garage, as shown in Figure 8, to cover all the areas in which the car could go through. The UWB anchors outside the garage (A0 to A3) were located with the same Septentrio GNSS system, applying Real-Time-Kinematics (RTK) corrections to the positioning algorithm, in order to obtain centimeter-level accuracy. The anchors inside the garage (A4 to A7) were located according to a local-frame X–Y–Z coordinate system, which origin was referenced to anchor A0. Moreover, the computation of the position of anchors A4 to A7 was performed by means of rotating the mentioned relative local X–Y–Z coordinates to East–North–Up (ENU) coordinates for a later translation to ECEF absolute coordinates. Note that any position error introduced in the reference A0 anchor's coordinates would directly translate to the coordinates of anchors A4 to A7. Accordingly, the RTK algorithm was allowed to converge over 15 min, in order to obtain centimeter-level accuracy.

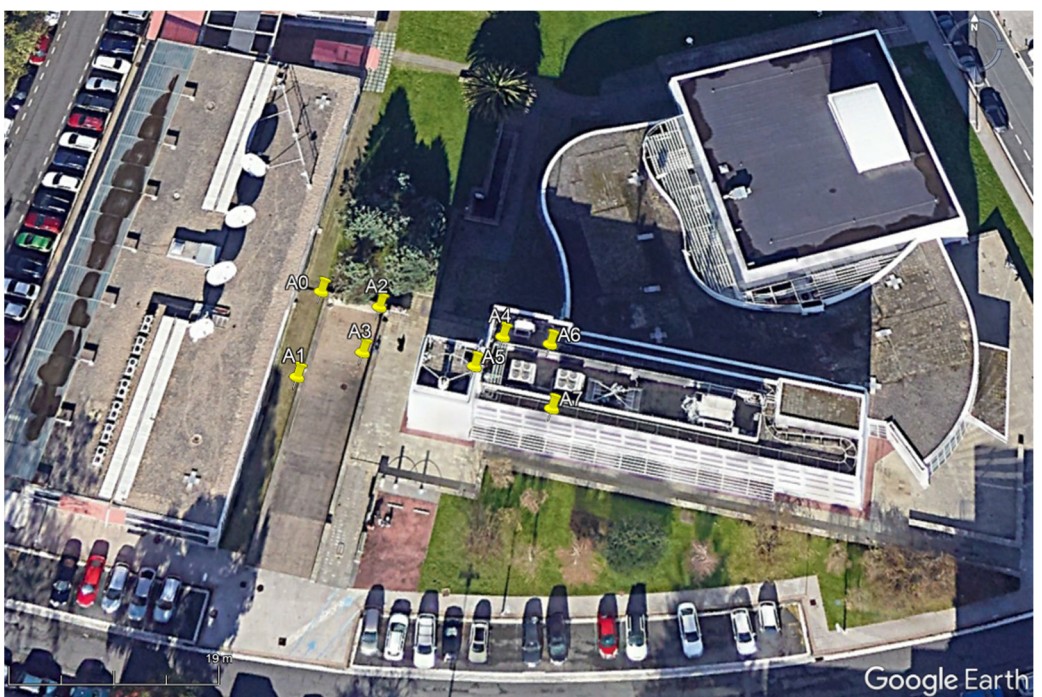

**Figure 8.** Distribution of the UWB anchors within the garage area.

For the relative location of said anchors, the accurate position of the rest of the anchors was used, as the solutions obtained from the RTK positioning were assumed to be accurate enough. The ground truth, on the other hand, was obtained using the same high-performance Septentrio GNSS receiver, applying the required corrections to obtain an RTK centimeter-level position.

### 3.3. Test Cases

In order to obtain the results shown in Section 4, seven different and sequential rounds were performed in the selected test site.

Each and every round started inside the coverage range of the UWB anchors, right outside the entrance of the garage. After letting the system converge for 10 to 15 s, the car left this area, decoupling from the UWB ground infrastructure. During the GNSS standalone phase of the track, the car went through the friendliest part of the performed track (environments 1 and 2 from Section 3.1), only to later proceed into the coverage of the UWB anchors, where the fusion between UWB and GNSS could be found once again. Inside this UWB coverage region, the car performed the same maneuver inside the garage (indoor environment, without GNSS visibility) in order to get back to the initial point.

Due to the fact that a 40° elevation mask was applied to GNSS observables, discarding the GNSS measurements that tend to be more vulnerable to faults, the FDE method is expected to have a higher probability of excluding measurements corresponding to UWB anchors.

## 4. Result and Discussion

This section shows and analyzes the results obtained when applying the employed FDE methods to the fusion between GNSS and UWB by means of a Kalman Filter. These results were obtained after processing the data collected in the measurements explained in the previous Section 3. The computation of the error was performed by comparing the calculated solution's position against the ground truth's coordinates with the closest temporal value. Since the output data rate of this ground truth was ten times higher than the one of the developed navigation algorithm, a maximum time error of 0.1 s could be suffered and, thus, added to the computed error. Due to the fact that UWB anchors were deployed just in the surrounding area of the garage (see Figure 8), the results analysis is divided into two main parts, being intuitively classified depending on the use of UWB as an augmentation system.

Accordingly, the two main regions of interest to be distinguished are the one in which UWB signals are received and the one in which they are not. Due to the fact that the 40° satellite elevation mask reduces the probability of finding a faulty GNSS observable (a fact that will be shown in later figures), the main region of interest for this research work is the one that comprises UWB signal coverage. As a consequence, the following result discussion will refer to the effects of the employed FDE method inside this coverage region.

The following Table 3 summarizes the results of the multiple measurement rounds performed in the test site described in the previous Section 3. This table evaluates each of the performed seven rounds in terms of the number of computed solutions and horizontal error, divided into the minimum, mean and maximum values, together with its variance. Note that, due to the high observable redundancy created by the number of visible satellites and the eight deployed anchors, the likelihood of reducing system solution continuity due to excessive exclusions or lack of redundancy was significantly reduced. Accordingly, only two epochs were declared unreliable, as seen in rounds 1 and 5.

**Table 3.** Obtained results over multiple measurements.

| | Navigation Method | | | | | | | | | |
| | GNSS+UWB | | | | | GNSS+UWB+FDE | | | | |
| Round | Number of Solutions | Error | | | | Number of Solutions | Error | | | |
| | | Min. | Mean | Max. | Variance | | Min. | Mean | Max. | Variance |
|---|---|---|---|---|---|---|---|---|---|---|
| 1 | 1938 | 0.25 | 2.08 | 10.14 | 5.93 | 1937 | 0.25 | 2.06 | 10.14 | 5.91 |
| 2 | 1707 | 0.22 | 2.58 | 17.99 | 9.08 | 1707 | 0.22 | 2.57 | 9.4 | 9.08 |
| 3 | 1658 | 0.04 | 2.02 | 15.49 | 5.76 | 1658 | 0.04 | 2.02 | 8.3 | 5.76 |
| 4 | 1409 | 0.24 | 2.71 | 17.64 | 11.21 | 1409 | 0.24 | 2.71 | 6.4 | 7.03 |
| 5 | 1913 | 0.06 | 1.77 | 10.64 | 5.4 | 1912 | 0.06 | 1.76 | 5.1 | 4.63 |
| 6 | 982 | 0.41 | 3.19 | 11.77 | 5.35 | 982 | 0.42 | 2.82 | 7.74 | 6.44 |
| 7 | 1042 | 0.23 | 2.93 | 12.54 | 4.27 | 1042 | 0.29 | 2.6 | 8.96 | 4.05 |

In the said table, it can be seen that, overall, applying the described FDE technique improves the performance of the GNSS + UWB positioning algorithm. Due to the fact that applying FDE especially focuses on the detection of outliers, it is only normal to observe

that the results showed an improvement in the maximum values of the error, which is then reflected in the means and the variances. Accordingly, the minimum error values did not significantly vary, as the minimum values usually corresponded to the best observable situations, observables which should not be excluded by the FDE methods; if they are, this may be due to a non-sufficient probability of the false alarm, $P_{fa}$.

Table 4 shows the effect of applying FDE methods to the UWB-augmented GNSS navigation, as it shows the percentage variation of the two halves of Table 3. This way, the reductions in the positioning error are colored in green, while the increases are colored in red. Accordingly, the improvement in the maximum positioning error becomes obvious, since it reached reductions up to 64%, which led to a decrease in the mean positioning error by up to 12%. Note that the reductions in the number of solutions are colored in yellow since these cannot be evaluated at the same time in terms of accuracy, integrity and continuity. Having said this, the reduction in the number of solutions induced a slight decrease in the continuity of the system as compensation for the improvement in the mean error, and a reduction of the maximum error.

**Table 4.** Percentage variation of the obtained results when applying FDE.

| Round | Number of Solutions | Improvement Percentage after Applying FDE | | | |
|---|---|---|---|---|---|
| | | Error | | | |
| | | Min. | Mean | Max. | Variance |
| 1 | −0.1% | 0% | −1% | 0% | 0% |
| 2 | 0.0% | 0% | 0% | −48% | 0% |
| 3 | 0.0% | 0% | 0% | −46% | 0% |
| 4 | 0.0% | 0% | 0% | −64% | −37% |
| 5 | −0.1% | 0% | −1% | −52% | −14% |
| 6 | 0.0% | 2% | −12% | −34% | 20% |
| 7 | 0.0% | 26% | −11% | −29% | −5% |

Furthermore, it is meaningful to mention that in the part of the track in which UWB was not used or, what is to say, the part outside the entrance of the garage, the results of the GNSS and UWB fusions with and without applying the FDE method performed almost identically. Note that, after losing the coverage of the UWB anchors, the GNSS + UWB and GNSS + UWB + FDE solutions need some time to converge to the same standalone GNSS solution.

Moreover, as shown in Figure 9, the result at the beginning of the track of the GNSS + UWB and GNSS + UWB + FDE methods are almost identical, being both of them much closer to the ground truth than the GNSS-only one, which showed an initial offset of about 4 m from the reference (see Figure 10). The cause of this is that the convergence time at the beginning of the measurement allows the algorithm to make use of UWB's accuracy to obtain an accurate initial position. Moreover, the similarity between the GNSS + UWB and GNSS + UWB + FDE methods can be confirmed by the number of excluded observables at the beginning of the measurement; which can be seen in Figure 11. It can be seen that the only observables that are excluded are the UWB ones due to two main reasons. The first one, as commented in the introduction of this document, is related to the elevation mask applied to the satellites in view which, as it is set to be higher than 40°, reduces the probability of employing multipath or NLOS-containing satellite pseudoranges. The second one is caused by the high confidence value that is assigned to UWB ranges by the measurement covariance matrix, $R_k$, due to their lower noise Power Spectral Density (PSD). This high confidence also implies higher visibility at the time of fault detection, making them easier to be detected, as expected.

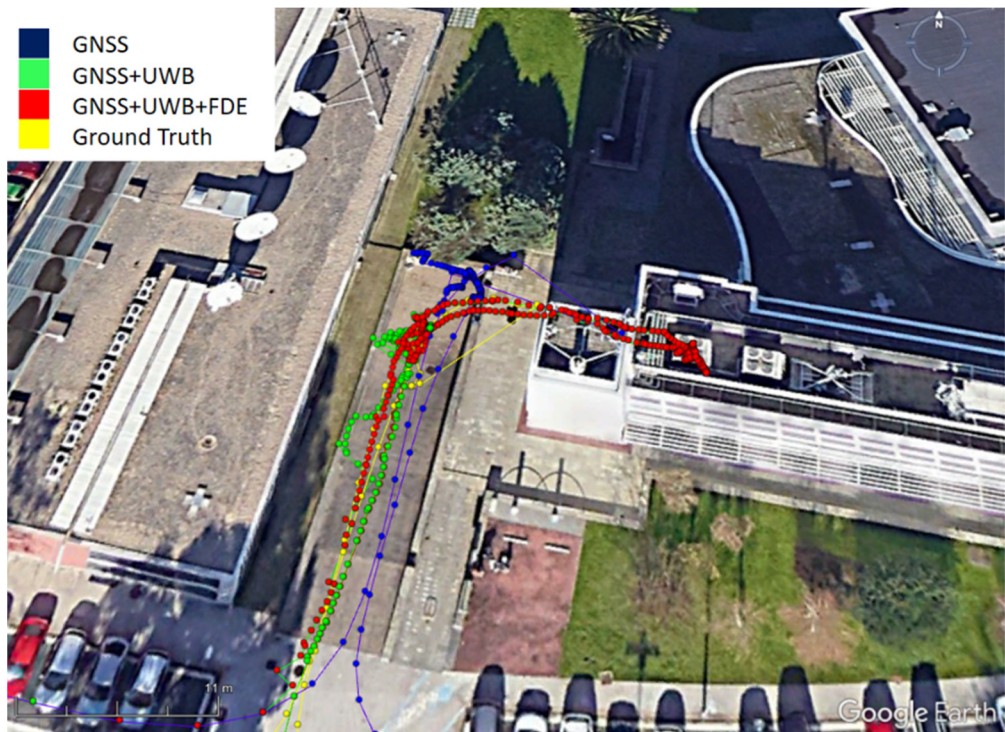

**Figure 9.** Qualitative comparison of the results against the ground truth (round 5).

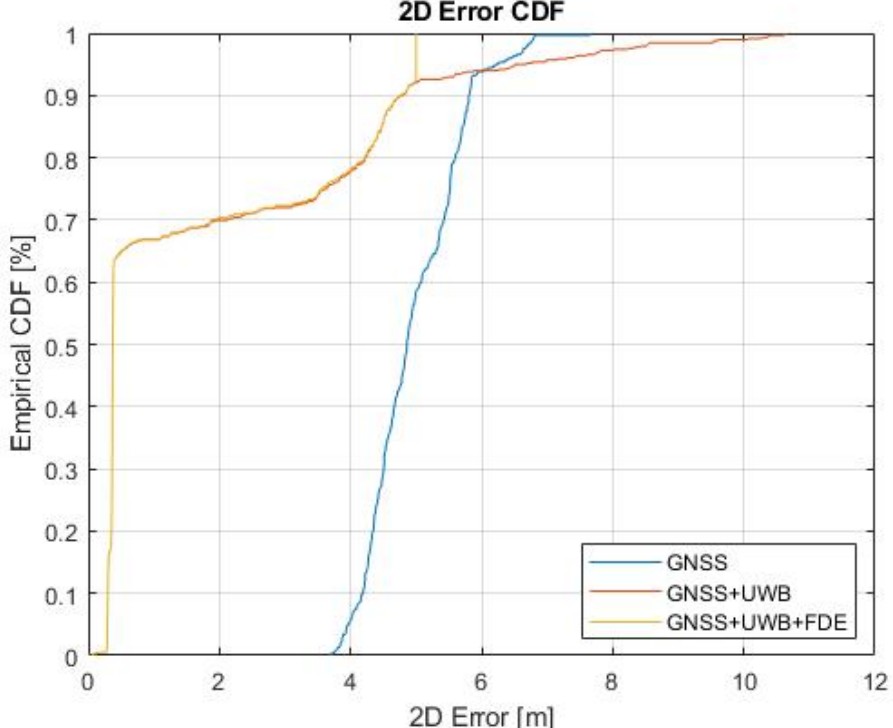

**Figure 10.** Comparison of the 2D error cumulative distribution functions (round 5).

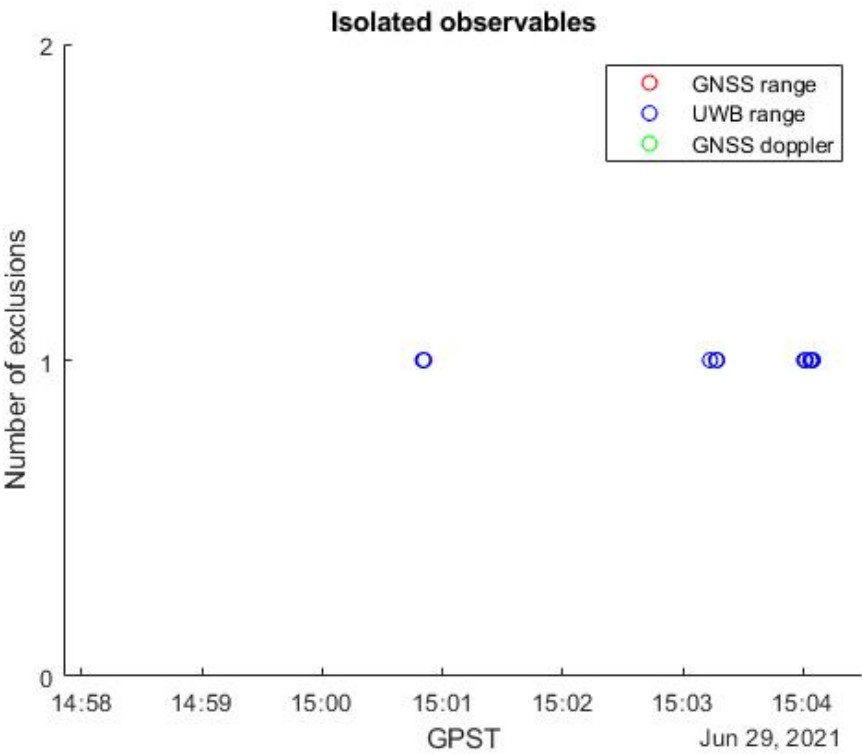

**Figure 11.** Number of exclusions for each type of measurement (round 5).

After losing sight of the last anchor and navigating through the outdoor part of the track, it is at the time of recoupling with the UWB anchors where the biggest differences in results can be found. As seen in Figure 9, the three methods (GNSS, GNSS + UWB and GNSS + UWB + FDE) did not show the same solution position at the beginning of the entrance of the garage, being the GNSS-only method further from the reference than the UWB-employing ones. Moreover, it is only meaningful to comment that, as happened at the start of the mission, the only UWB anchors in view were the ones from A0 to A5, which is intuitively expectable due to the proximity and the geometry of the garage. However, the weak reception of the signal coming from anchor A5, which may be caused by the lack of line-of-sight reception, is what makes the difference between using and not using FDE methods to obtain higher accuracy and integrity.

This single fault isolation seen in Figure 11 of the range incoming from anchor A5 causes the qualitative improvement that can be seen in Figure 9 (removal of the green divergence at the middle of the garage entrance and the removal of the green noisy cloud at the end of the measurement). Moreover, this improvement is quantitatively observed in Figure 10 and round 5 in Table 3, as the cumulative distribution function of the FDE-using method limits its maximum value to 5.1 m, which is considerably lower than the 10.64m error shown by the non-FDE GNSS + UWB fusing method.

Moreover, not only does Figure 10 show the overall improvement of the UWB-augmented GNSS system against the standalone GNSS navigation, but it also shows that the degradation in performance of said sensor fusion is caused by incorrect UWB signal couplings or uncouplings, which are modelled as biased observables, thus, excluded by the FDE methods. Accordingly, the maximum positioning error is decreased, improving the behavior of the CDF that corresponds to GNSS + UWB + FDE with respect to the one corresponding to GNSS + UWB.

When analyzing Figure 12, which shows the visibility of each anchor during round 5 of the measurement campaign, a coherent behavior can be observed. To begin with, since an initial convergence time was awaited at the initial point of the track (right in front of the garage), a stationary phase can be observed in the UWB range estimations. Note that in

this very point, Line-Of-Sight signals could be read at the UWB tag, which is represented by the eight observables scatter lines.

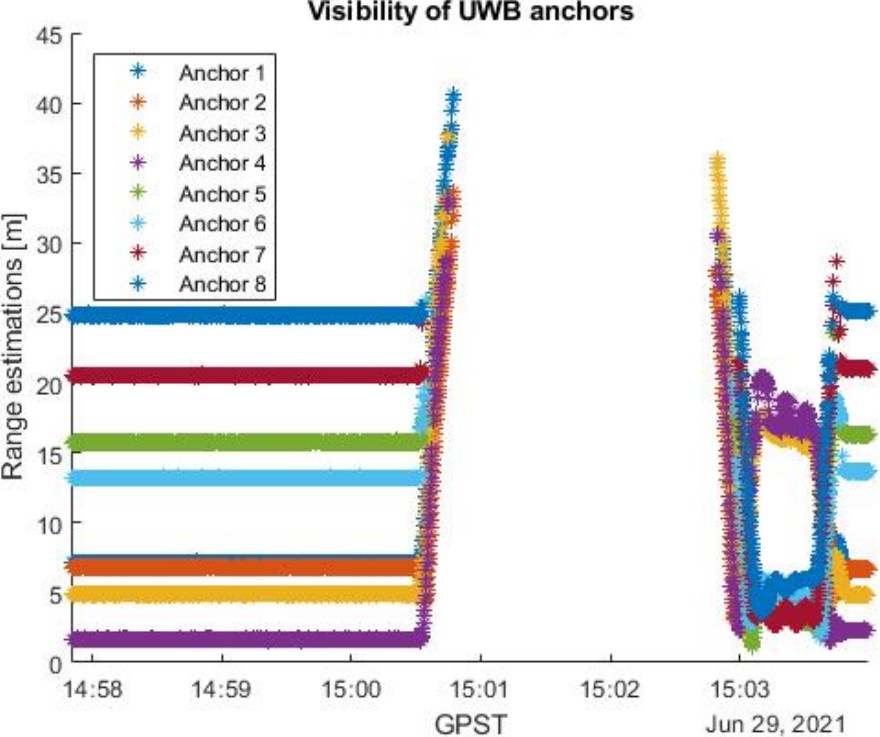

**Figure 12.** Visibility of UWB signals (round 5).

It is only when the vehicle starts its mission that the UWB range estimations began to change their values and disappear due to the lack of LOS of NLOS signals that the first excluded observable can be seen in Figure 11. This excluded range from anchor 5 is caused by the lack of LOS component in the received signal before it disappeared at the same time that the vehicle abandoned the UWB coverage area.

Analogously, no more UWB range exclusion could be found until the moment in which the car proceeded into the UWB coverage region, a moment in which no LOS component can be received from the signal incoming from anchor 5. Once again, this anchor seems to be close enough to receive its NLOS signal components, but it is not located in an appropriate location in order to receive its LOS signal component.

## 5. Conclusions

This paper confirmed the validity of Fault Detection and Exclusion methods to discard faulty UWB observables when these are used as an augmentation system for GNSS in urban and indoor environments. Accordingly, it was proven that the FDE methods that were originally designed for standalone GNSS navigation can be applied to range-based sensor-augmented GNSS systems when fusing said technologies by means of a Kalman Filter, which implies an advance from the first part of this research shown in [16].

According to the results in Table 3 and Figure 9, applying FDE methods to a UWB-augmented GNSS system provides a smoother and more accurate result since it removes not only faulty UWB observables but also divergent solutions. Therefore, it is a useful way to reduce the biggest errors in the solution by up to 64%, reducing the variance by up to 37% and slightly improving the mean error by up to 11%, as seen in Table 4. Moreover, these methods have been proven to be useful to remove faulty UWB augmentation observables when these are biased due to interferences such as multipath.

Lastly, applying FDE methods to the UWB augmented GNSS navigation allows a reduction of the positioning error of a low-cost GNSS up to decimeter level and can be used

to provide a navigation system with continuity in indoor environments, as it ensures the capability of UWB observables to improve low-cost GNSS systems in terms of accuracy, integrity and continuity.

**Author Contributions:** Conceptualization, P.Z.; methodology, P.Z.; software, P.Z.; validation, P.Z.; formal analysis, P.Z.; investigation, P.Z.; resources, P.Z.; data curation, P.Z.; writing—original draft preparation, P.Z.; writing—review and editing, G.D.M., J.M. and I.A.; visualization, G.D.M., J.M. and I.A.; supervision, G.D.M., J.M. and I.A.; project administration, J.M.; funding acquisition, J.M. All authors have read and agreed to the published version of the manuscript.

**Funding:** This research received no external funding.

**Data Availability Statement:** Not applicable.

**Acknowledgments:** The authors would like to thank the AutoEv@l project, in which this research work has been performed.

**Conflicts of Interest:** The authors declare no conflict of interest.

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
