# Peer review of "Innovation-Based Fault Detection and Exclusion Applied to Ultra-WideBand Augmented Urban GNSS Navigation"

_remotesensing, doi:10.3390/rs15010099_

Round 1

Reviewer 1 Report

The manuscript reported the performance improvement caused by the application of Fault Detection and Exclusion methods when applied to a UWB-augmented low-cost GNSS system in urban environments. Typically, GNSS provides a worldwide absolute outdoor positioning and becomes a reference technology in terms of navigation technologies. The performance of GNSS-based navigation gets degraded when employed in urban areas in which satellite visibility is not good enough or nonexistent. UWB technology can be a perfect candidate to complement GNSS as a navigation solution. Nevertheless, this fusion is vulnerable to interferences affecting both systems. The aim of this work is to solve the issues of the misbehaviour of an augmentation system, which may lead to unexpected and critical faults instead of improving the performance of the standalone GNSS.

I consider the content of this manuscript meets the reading interests of the readers of the Remote Sensing journal. However, there are certain English spelling and grammar issues, and also the discussion and explanation should be further improved. I suggest giving a minor revision and the authors need to clarify some issues or supply some more experimental data to enrich the content. This could be comprehensive and meaningful work after revision.

1. For grammar issues, it is suggested that the author double-check the small grammar errors in the full text, especially the lack of and redundant use of definite articles.

2. For the Keywords, Accuracyand Integrity should be omitted.

3. Page 2, ‘The use of Ultra-Wideband (UWB) radio technology in urban environments is a reliable option when trying to solve the main drawback of GNSS in low-visibility scenarios What are the common methods used to solve the low visibility problem of GNSS? There should be a brief overview of various methods, such as IMU, WIFI, or visual sensor. Later, the authors can emphasize the advantages of UWB. Why does it stand out from a variety of methods and become the focus of this manuscript?

  In the current piece of work, the performance-improving effect of said methods is shown when applied to the innovation vector of a Kalman Filter when this is used to employ UWB signals as an augmentation for GNSS navigation. The necessity of selecting the Kalman filter is not described and discussed enough. To be honest, since the system model is hard to characterize, while the process and measurement noise variances are also hard to find, I do not think only the Kalman filter is enough. Moreover, what has to be mentioned is that besides the Kalman filter, the recently improved Kalman filter should also be discussed. Please see [IEEE Transactions on Automatic Control, 67 (7) (2022) 3458–3471] and [Remote Sensing 12.4 (2020): 732.]. Including some discussion may make the literature review more comprehensive.

4.  Page 3, A precise definition of all the model matrices should be given, including the transition matrix, the covariance matrix of the system noise and measurement matrix.

5. Page 4, This diagonal matrix is open to characterization, as different approaches can be found in the literature. What are the different approaches exactly as reported in the literature? These methods should be simply listed or simply classified, and relevant literature should be appropriately cited. Otherwise, the description of this sentence does not bring readers more effective information.

6. On Page 6, the resolution of Figure 2 is very low, which should be changed to a high-resolution one. The same applies to Figure 3 on Page 7.

7. Page 7, Moreover, the likelihood of finding a faulty observable becomes more notorious when the observables are used due to the combination of GNSS and UWB technologies. The meaning of the word notorious is not very clear, the possibility is higher or not should be clarified.

8. Page 11, 4. Result discussion should be 4. Result and discussion. the FDE method is expected to have a bigger impact on the measurements corresponding to UWB anchors. The bigger impact is not very clear, it is the better impact or worse impact should be clarified

Table 2, the Variance on the right is not shown completely. Table 3 seems like a figure with very low resolution, not a real Table.

9. The description of the conclusion does not fully summarize the main results of this manuscript, especially since the summaries are mainly qualitative descriptions. I suggest adding some quantitative concrete results.

Author Response

The point-by-point answers to the reviewer's comments can be found in the following document. Some of the comments were very constructive.

Reviewer 2 Report

This paper was a pleasure to read and I think it is worthy of publication with a few changes.

1. I think the paper will be interesting to those seeking to implement similar ideas in their own research. That is to say there is a reader market who will benefit.

  2. The standard of English expression is very good overall, but some sentences were turgid or difficult to understand. For example, the sentence starting at the middle of line 100 is poorly constructed and needs improvement. This is one of several examples. It’s clear the authorship includes some good writers (of English) and a pass over the paper by the author team to improve clumsy sentences would improve the final outcome.

3. The paper could be improved by better referencing prior work. For example, the paper doesn't reference the equation set used to implement the Kalman filter; there is providence of the KF itself but more importantly notation varies, and it can be helpful to those looking to use the work to where the paper sits in the KF notation universe.

There are other references that should be made.

·       Statistical testing on innovation sequences goes back to work in the 1970s (Mehra and Peshcon),  whilst Willsky’s work in the mid-to-late 1970s using the statistics of innovation sequences for fault detection and identification shaped a lot of thinking here, particularly for local model tests. These are probably worth referencing for providence of thinking.     

·       Teunissen and Salzmann’s work in the late 1980s and early 1990s is generally acknowledged to have introduced quality control theory based on the standard state space Kalman filtering model and the statistical hypothesis testing into geodetic surveying. This work seems particularly relevant because it references local model tests and global test. The statistical tests bear some connection.

·       The use pseudorange, carrier phase, and phase rate (Doppler) observations in a Kalman filter to determine the instantaneous position, velocity and even acceleration are set down in well-known references such as Schwarz et al,1989; Cannon,1990; Hwang and Brown,1990.

4. I've done work with UWBs for pose determination within a KF strucuture and found one of the challenges is the non-synchronous nature of the range measurements they provide.  I don't know if the units used here had this property, but regardless it’s not clear in the formulation how different measurement times were accommodated within the Kalman filter structure. UWB range outlier detection for a system in motion will be influenced by timing of measurements, and increasingly so as the speed of the platform increases.  Some detail of these challenges and how they are solved would be useful.

4. There are no details of the process model (that I could find). I expect it is constant velocity or constant acceleration of the antenna center point. Some details would be useful, again with a reference for the reader to go to. There are many, but one aligned to the notation used would help.

5. Global Test statistics are done on an epoch of innovations. There is no indication the of this of the epoch period in the description of the test statistics.

6. The GNSS and UWB antenna do not share the same centres (from Figure 7 they look to be 20-30cm apart) and so I would expect the process model measurement equations would need to account for this. I haven’t written them out but I expect they will be non-linear. Some detail might be useful for anyone repeating the work including whether the estimator is an extended Kalman filter. 

7. The UWB ad GNSS chipset used for trials are given: Ublox M8T GNSS received and DW1000 based UWB systems. I think the paper would benefit from short specification table (or similar) for this hardware giving pertinent info.  I found myself having to look the specifications up. Not a big task but a distraction in reading the paper. I expect other readers would also benefit from such a table for the same reasons.

Overall, the usefulness of the paper is summarized well by Figure 10. I expect that the CDFs for the three 'algorithms' will be dependent environment geometry so this is really an example for a specific environment, but the result is powerful none the less.  Whilst the paper might be criticised as an obvious adaption of a standard approach, the contribution is not so much the ideas underpinning the work, but the demonstration that range measurements from other sources can be incorporated and that outliers in these measurements can be managed by statistical tests on the innovation sequence to improve performance.

I found the paper very thought provoking and I intend to try some things as a consequence.  I expect others will react similarly.  

Author Response

(The authors gave the same response as above.)

Reviewer 3 Report

The authors present an innovative fault detection algorithm for improving the combined GNSS UWB positioning; however, the improvement is practically negligible, see Table 2. 

The paper presents much information on a high level, so the paper review is problematic. 

1. The information about the estimated parameters vector x is missing, so the reviewer cannot check how or whether the authors model the GNSS receiver clock, for example. 

2. The information about the positions of the UWB anchors is missing. The anchor geometry has a cardinal impact on the precision.

3. The system requires local elements that can distribute additional data like DGNSS corrections. The DGNSS corrections can improve the precision of the GNSS pseudoranges. The information on the implementation of differential measurement is missing as the analyses of their impact.

4. In Table 2. the authors analyze the measurement errors. It is not clear how the authors determine those errors. It is not and trivial task. 

The authors should add the missing information. A major revision is needed. 

Author Response

The point-by-point answers to the reviewer's comments can be found in the following document.

Round 2

Reviewer 3 Report

The reviewer comments have been accepted, and the manuscript has been updated.